# Beyond the Warburg Effect: Oxidative and Glycolytic Phenotypes Coexist within the Metabolic Heterogeneity of Glioblastoma

**DOI:** 10.3390/cells10020202

**Published:** 2021-01-20

**Authors:** Tomás Duraj, Noemí García-Romero, Josefa Carrión-Navarro, Rodrigo Madurga, Ana Ortiz de Mendivil, Ricardo Prat-Acin, Lina Garcia-Cañamaque, Angel Ayuso-Sacido

**Affiliations:** 1Faculty of Medicine, Institute for Applied Molecular Medicine (IMMA), CEU San Pablo University, 28668 Madrid, Spain; tom.duraj.ce@ceindo.ceu.es; 2Faculty of Experimental Sciences, Universidad Francisco de Vitoria, 28223 Madrid, Spain; noemi.garcia@ufv.es (N.G.-R.); pepa.carrion@ufv.es (J.C.-N.); rodrigo.madurga@ufv.es (R.M.); 3Brain Tumor Laboratory, Fundación Vithas, Grupo Hospitales Vithas, 28043 Madrid, Spain; 4Fundación de Investigación HM Hospitales, HM Hospitales, 28015 Madrid, Spain; aomendivil@yahoo.es; 5Neurosurgery Department, Hospital Universitario La Fe, 46026 Valencia, Spain; ricprat@hotmail.com; 6Departamento de Medicina Nuclear, HM Hospitales, 28015 Madrid, Spain; lgarciacanamaque@hmhospitales.com

**Keywords:** glioblastoma, energy metabolism, glycolysis, oxidative phosphorylation, therapeutics, gene expression profiling

## Abstract

Glioblastoma (GBM) is the most aggressive primary brain tumor, with a median survival at diagnosis of 16–20 months. Metabolism represents a new attractive therapeutic target; however, due to high intratumoral heterogeneity, the application of metabolic drugs in GBM is challenging. We characterized the basal bioenergetic metabolism and antiproliferative potential of metformin (MF), dichloroacetate (DCA), sodium oxamate (SOD) and diazo-5-oxo-L-norleucine (DON) in three distinct glioma stem cells (GSCs) (GBM18, GBM27, GBM38), as well as U87MG. GBM27, a highly oxidative cell line, was the most resistant to all treatments, except DON. GBM18 and GBM38, Warburg-like GSCs, were sensitive to MF and DCA, respectively. Resistance to DON was not correlated with basal metabolic phenotypes. In combinatory experiments, radiomimetic bleomycin exhibited therapeutically relevant synergistic effects with MF, DCA and DON in GBM27 and DON in all other cell lines. MF and DCA shifted the metabolism of treated cells towards glycolysis or oxidation, respectively. DON consistently decreased total ATP production. Our study highlights the need for a better characterization of GBM from a metabolic perspective. Metabolic therapy should focus on both glycolytic and oxidative subpopulations of GSCs.

## 1. Introduction

Glioblastoma (GBM) is the most common, heterogeneous and aggressive primary brain tumor in adults (54% of all gliomas) [1,2,3]. The World Health Organization (WHO, Geneva, Switzerland) classifies GBM based on histopathological findings and molecular features (especially IDH mutation status) [4]. At a gene-expression level, GBM can be classified into four subtypes: mesenchymal, classical, proneural and neural [5,6].

Standard treatment of GBM consists of maximally safe surgical resection, followed by radiotherapy and chemotherapy, usually in the form of temozolomide (TMZ). Despite decades of extensive research and advancements in therapeutics, such as tumor treating fields (TTF), prognosis remains extremely poor, with a median overall survival of 20.9 months [7]. GBM has a low global incidence (less than 10 per 100,000 persons/year), but cumulative survival after five years from diagnosis is less than 10%, making it a critical public health issue [8,9]. Dismal survival is partly owed to GBM’s highly invasive, chemo-resistant and recurrent nature [10]. As standard of care is not a curative option, new therapies are sorely needed, with efforts to characterize GBM from multiple viewpoints, predominantly the omics sciences.

Setting aside the uncertainty behind the origin of cancer [11,12], one of its defining characteristics, at a functional, bioenergetic level, is its ability to exploit glycolytic metabolism even in the presence of oxygen, a phenomenon known as the “Warburg effect” [13]. Among many other solid tumors, this metabolic shift has been extensively documented in gliomas [14,15]. In the mitochondrial theory of cancer, aerobic glycolysis represents a universal feature of transformed cells, allowing the reduction of vast molecular heterogeneity into a smaller number of metabolic categories [16]. Metabolic reprogramming is not merely an in vitro artifact, but has wide-ranging clinical applications [11,17,18]. Nowadays, 18F-fluorodeoxyglucose PET (18F-FDG PET) is a common technique for cancer diagnosis and staging, with novel metabolic markers, such as lactate, glutamine, oxygen and even ketone bodies under clinical evaluation [19,20,21]. Within ample cell diversity, however, the predominance of aerobic glycolysis does not necessarily abrogate ancillary energetic sources: functional oxidative metabolism (glucose, fatty acids, glutamine) and the “reverse Warburg” effect [22,23]. Characteristically, in vitro, GBM has shown high variability in mitochondrial respiration, while tissue-derived cell lines revealed glucose dependency and fatty acid oxidation (FAO) [23,24,25,26]. Intratumoral heterogeneity makes development of targeted strategies against specific mutations very challenging [27]; therefore, patient stratification based on metabolic pathways should be a key component of improved therapeutic strategies.

The main challenge facing GBM management is the eradication of all malignant cells, including those able to survive drastic changes in the tumor microenvironment and toxic interventions. For this reason, GBM presents as a unique model to study bioenergetic alterations, as both aerobic glycolysis and oxidative phosphorylation (OXPHOS) have been described in high grade gliomas [28,29,30]. Furthermore, discouraging survival rates are a compelling reason to explore new therapeutic opportunities, either stand-alone or, more likely, in combination with standard of care. To this effect, metabolic inhibitors such as metformin hydrochloride (MF), dichloroacetate (DCA), sodium oxamate (SOD) and 6-diazo-5-oxo-L-norleucine (DON) have a longstanding history in this field, undergoing extensive evaluation in animal models and clinical trials with a variable rate of success [31,32,33,34].

To accurately model this disease in vitro, it has been proposed that GBM stem cells (GSCs) have a remarkable proliferative ability, sufficient to drive tumor maintenance, recurrence and therapeutic resistance [35,36,37,38]. GSCs are a highly heterogeneous and metabolically adaptive cell population: surviving in both perivascular aerobic and hypoxic regions [39,40], seemingly able to shift between glycolytic and oxidative phenotypes [28,29]. Whether these parameters are permanent, stable, independent or complementary, operating on a spectrum, remains to be elucidated [41].

To help us illuminate this question, we performed a Gene Set Variation Analysis (GSVA) [42] for canonical glycolytic/oxidative pathways in The Cancer Genome Atlas (TCGA) GBM datasets. A clustering of highly oxidative signatures was observed in normal tissues, whereas highly glycolytic tumors matched with the mesenchymal subtype; interestingly, mesenchymal signatures are associated with increased inflammation and wound healing pathways, a higher degree of necrosis and the worst survival when restricting for samples with low transcriptional heterogeneity [5,43]. Between these two categories, a high degree of heterogeneity was recognized. Clinically, 18F-FDG PET imaging of GBM can exhibit high or low glucose uptake, but allocation of metabolic substrates is not routine practice.

To verify these observations in vitro, we analyzed the basal metabolic phenotype of three tissue-derived and molecularly distinct GSCs (GBM18, GBM27, GBM38), in addition to traditional established cell line U87MG. As metabolic plasticity is being touted as a distinctive feature of GSCs, we wanted to explore antiproliferative responses to metabolic inhibitors and their correlation with basal bioenergetics. High resistance to MF (a mild mitochondrial inhibitor) and DCA (glycolytic modulator) was detected in GBM27, a distinctively oxidative cell line. GBM38 displayed Warburg-like properties, with higher sensitivity to DCA. Responses to DON (glutaminase inhibitor) varied between cell lines, without a clear correlation with basal metabolic phenotypes. Subsequently, we combined promising drug candidates with bleomycin, a radiomimetic drug that causes single-strand and double-strand DNA breaks [44,45]. Synergism at therapeutically relevant outcomes was detected with all drugs in GBM27, and all cell lines with DON. Lastly, Seahorse XF analysis was performed in surviving, metabolically treated cells to determine vulnerabilities in bioenergetic phenotypes (“metabolic priming”).

Here, we propose that strategic targeting of dysregulated bioenergetic pathways, after an initial assessment of the metabolic phenotypes coexisting within a tumor, could become a valuable stratification and therapeutic tool, improving the efficacy of adjuvant metabolic therapy.

## 2. Materials and Methods

### 2.1. Culture of GSCs from Human GBM Samples, U87MG and Mesenchymal Stem Cells

GSCs were originally isolated from surgical human GBM specimens, as described by our group in [46]. The GSCs used in this study are characterized by distinct molecular and morphological features, differential drug sensitivity profiles and in vivo dissemination patterns that reflect the original tumors. GSCs were cultured under a humidified atmosphere of 5% CO_2_ at 37 °C, in a media containing, as a base, DMEM/F-12 (catalog number 11039, Gibco, Grand Island, NY, USA), supplemented with: Non Essential Amino Acids (1% *v/v*; 11140, Gibco), HEPES (38 mM; 15630, Gibco), D-Glucose (0,54% *v/v* or 30 mM; G8769, Sigma-Aldrich, St. Louis, MI, USA), BSA-FV (0,01% *v/v*; 15260037, Invitrogen, Carlsbad, CA, USA), Sodium Pyruvate (1 mM; Invitrogen), L-Glutamine (4 mM; 25030, Gibco), Antibiotic-Antimycotic (0.4% *v/v*; Invitrogen), N1 Supplement (1% *v/v*; Invitrogen), Hydrocortisone (0.3 μg/mL; H0135, Sigma-Aldrich), Tri-iodothyronine (0.03 μg/mL; T5516, Sigma-Aldrich), EGF (10 ng/μL; E9644, Sigma-Aldrich), bFGF (20 ng/mL; F0291, Sigma-Aldrich) and Heparin (2 μg/mL; H3393, Sigma-Aldrich).

U87MG was purchased from ATCC, Rockville, MD, USA and cultured in DMEM/F-12 (11039, Gibco) supplemented with 10% fetal bovine serum (FBS) and 2% penicillin-streptomycin (PS). Cells were maintained at 37 °C in humidified atmosphere air, CO_2_ 5%.

Human mesenchymal stem cells (hMSCs) (a gift from Dr. Carmen Escobedo Lucea) were cultured in DMEM, high glucose, GlutaMAX (10566016, Gibco), supplemented with a final concentration of 20% FBS and 1% P/S. All hMSCs experiments were performed in the first five passages from isolation.

### 2.2. Reagents and Metabolic Inhibitors

1,1-Dimethylbiguanide hydrochloride (D150959), sodium oxamate (O2751), sodium dichloroacetate (347795) and 6-Diazo-5-oxo-L-norleucine (D2141) were purchased from Sigma-Aldrich. Bleomycin sulfate (HY-17565) was acquired from MedChemExpress, Monmouth Junction, NJ, USA.

### 2.3. MTS Assays and Drug Combination Studies using the Chou-Talalay Method

The sensitivity to different metabolic drugs was assessed using [3-(4,5-dimethylthiazol-2-yl)-5-(3-carboxymethoxyphenyl)-2-(4-sulfophenyl)-2H-tetrazolium, inner salt (MTS) containing solution from Promega, Madison, WI, USA (CellTiter 96 AQueous One Solution, G3582). Briefly, single-cell suspensions of GSCs were plated in a 96-well plate, 3000 cells/well, and allowed to grow and form spheres for 72 h. U87MG were seeded at 3000 cells/well and allowed to grow for 24 h. Cultures were then treated with their respective culture media (control cells) or increasing concentrations of each drug for 0 h, 24 h, 48 h or 72 h. At each timepoint, MTS reactant was added, incubated at 37 °C for 2 to 4 h and absorbance was measured at 490 nm/630 nm, using a Varioskan Flash (5250030, Thermo Scientific, Waltham, MA, USA) or a Sunrise Absorbance Reader (Tecan Trading AG, Männedorf, Switzerland). For IC50 calculations, corrected absorbance was transformed, normalized and extrapolated in GraphPad Prism version 8.0.1, using the logarithmic variable slope equation:Y = 100/(1 + 10^((LogIC50 − X) × HillSlope)).

In specific dose experiments, hMSCs were seeded at 6000 cells/well and allowed to grow for 72 h. Fresh cell culture media was then added to control wells and dissolved treatments to experimental wells. Cells were treated for 72 h before MTS read-out.

Combinatory studies were performed in the same manner as single-drug assays. After seeding and cell-specific recovery/attachment intervals, combined treatments were added in the following final concentrations: IC50 for drug A alone; IC50 for drug B alone; full dose IC50 for drug A + drug B; IC50(A + B)/2; IC50(A + B)/8. Experimentally, drug “A” was one of the metabolic inhibitors (MF, DCA or DON), whereas drug “B” was the radiomimetic bleomycin. CompuSyn software (version 1.0), based on the Chou-Talalay method, was employed to determine the interaction between the drugs [47,48]. This method utilizes a multiple drug-effect equation derived from enzyme kinetics, generating a “combination index” (CI) for each drug combination, at each fraction of affected cells (Fa) level. CompuSyn software defines synergy as a CI value lower than 1, CI = 1 equals to additive effects and CI values > 1 indicate antagonistic effects. We have determined CI values for each metabolic inhibitor and bleomycin across all tested cell lines using a constant ratio experimental design, as well as other valuable parameters such as the Dose-Reduction Index (DRI), which indicates how many folds of dose-reduction for each drug, at any given effect, would be allowed in synergistic combination.

### 2.4. Real-Time Quantitative Reverse Transcription PCR (RT-qRT-PCR) Analysis

For RT-qRT-PCR, total RNA was isolated from cell pellets using NZYol (MB18501, NZYTech, Lda.), following the manufacturer’s recommendations. For chronological parity with other experiments, GSCs were seeded in 6-well plates at a density of 90000 cells/well, allowed to grow for 72 h, fresh cell culture media was added (1:1) and pellets were collected after 72 h; the same protocol was applicable to U87MG, but fresh cell culture media was added after 24 h from seeding. Purity of RNA was assessed based on 260/280 and 260/230 ratios using a Thermo Scientific NanoDrop 2000/2000c. RNA was retrotranscribed to cDNA (High-Capacity cDNA Reverse Transcription Kit; Applied BioSystems). Resulting samples were amplified with specific primers (Table 1) in a CFX Connect Real-Time PCR Detection System (Bio-Rad). *β-actin* and *GAPDH* were used as housekeeping genes. For relativization and comparison with a non-tumoral control, we compared our samples with a pool of retrotranscribed RNA from brain tissue obtained from epileptic patients, provided courtesy of Hospital Universitario y Politécnico La Fe (Valencia).

### 2.5. Antibodies

All primary and secondary antibodies were purchased from commercial sources, listed as follows: AMPKα Antibody (2532, Cell Signaling, Danvers, MA, USA), phospho-AMPKα (Thr172) (2535, Cell Signaling), Anti-Pyruvate Dehydrogenase E1-alpha subunit antibody (ab110334, Abcam, Cambridge, UK), Anti-PDHA1 (phospho S293) antibody (ab177461, Abcam), β-Actin (A5441, Sigma-Aldrich), α-Tubulin (sc-8035, Santa Cruz Biotechnology, Santa Cruz, CA, USA). The secondary antibodies for horseradish peroxidase (HRP) detection were anti-rabbit IgG (sc-2004, Santa Cruz Biotechnology) and anti-mouse IgG (PI-2000, Vector Laboratories, Burlingame, CA, USA).

### 2.6. Protein Isolation/Quantification and Western Blotting

Centrifuged and pelleted U87MG and GSCs were resuspended in 100 μL of radioimmunoprecipitation buffer [RIPA; 100 mM Tris-HCl (pH 8.5), 200 mM NaCl, 5 mM EDTA and 0.2% SDS, with phosphatase and a protease inhibitor cocktail and stored at −80 °C for a minimum of 24 h. Samples were then centrifuged at 13,200 RPM for 20 min at 4 °C; protein-containing supernatant was conserved.

Total protein concentration was determined using Bio-Rad Protein Assay according to the manufacturer instructions; after corresponding incubation, absorbance was read at 595 nm.

In phosphorylation experiments, treatments dissolved at 1:1 concentration in serum-free medium were added 3–4 days after seeding GSCs, and 24 h in the case of U87MG. U87MG cells were washed twice with PBS and serum-deprived for 1 h prior to sample collection. Protein was subsequently recovered at the indicated timepoints (30 min, 60 min, 2 h, 6 h).

Western blotting experiments were performed adapting the protocol from Mahmood et al. [49]. Briefly, protein extracts were separated by 8%–12% SDS-PAGE and transferred to nitrocellulose membranes. After blocking for 1 h with 5% Bovine Serum Albumin (BSA) in Tween-Tris Buffered Saline 1× [T-TBS; 10 mM Tris-HCl (pH 7.6), 150 mM NaCl and 0.1% Tween-20], membranes were incubated with the corresponding primary antibody O/N at 4 °C. After washing three times for 10 min with T-TBS, membranes were incubated with HRP-linked secondary antibody for 1 h at room temperature (RT). Detection was performed using ECL reagents (GE Healthcare) according to the manufacturer’s guidelines and revealed in a BioRad ChemiDoc chemiluminescence system. The same membranes were then incubated with a housekeeping primary antibody O/N at 4 °C, washed the next day and incubated with an HRP-linked secondary antibody for 1 h RT before ECL detection.

### 2.7. Seahorse XFp Protocol for Real-Time Metabolic Evaluation of U87MG Adherent Cells and GSCs Neurospheres

Experiments were performed in an XFp 8-well microplate using the Seahorse XFp Analyzer (Agilent, Santa Clara, CA, USA). Briefly, GSCs were seeded at a density of 10,000 cells/well and allowed to grow for 72 h, in wells previously coated with 20 µL of Collagen Type IV at 20 µg/mL (C6745-1ML, Sigma Aldrich). U87MG cells were seeded at 6000 cells/well and allowed to grow for 24 h. Metabolic drugs were added to the treatment wells and fresh media was added to the control wells. After 72 h, the original media was carefully pipetted out of each well into a centrifuge tube without disturbing the attached cells; then, Seahorse XF DMEM medium, pH 7.4 (103575-100, Agilent) was used to wash, pipetted out and centrifuged with the original media at 1000 rpm for 5 min at 25 °C. After centrifugation, supernatant was aspirated from each tube, conserving only the cell pellet, resuspended in Seahorse medium and added back to respective wells.

We then followed the standard protocol for Standard XF Real-Time ATP Rate Assay, as described in the Seahorse XF Real-Time ATP Rate Assay User Guide (Kit 103592-100, Agilent). Seahorse XF technology measures two key parameters of cellular bioenergetics: oxygen consumption rate (OCR; an estimation of mitochondrial ATP) and extracellular acidification rate (ECAR; quantification of glycolytic activity through changes in pH by lactate production) [50,51]. Results were analyzed in Seahorse Wave software (version 2.6.1), with analysis of OCR and ECAR carried out using the Seahorse XF Real-Time ATP Rate Assay Report Generator (version 4.0.17). For normalization, total protein was quantified using an Invitrogen Qubit 3 Fluorometer (Q33216, Invitrogen).

### 2.8. TCGA Gene-Set Variation Analysis

Affymetrix (HG-U133A) normalized gene expression datasets of GBM and non-tumor tissue samples from TCGA were downloaded from GlioVis repository (gliovis.bioinfo.cnio.es) [52]. As IDH mutation status confers characteristic metabolic rewiring of the TCA cycle, IDH mutant and IDH unknown samples were removed from the analysis [53]. The remaining 498 GBM IDH-wt and 10 non-tumor samples were classified in proneural, classical, mesenchymal and those with a high content in non-tumoral tissue (low cellularity), as proposed elsewhere [54]. Four different canonical gene sets (two oxidative and two glycolytic) were obtained from the Molecular Signatures Database (MSigDB) [55]: KEGG (oxidative phosphorylation and TCA cycle), Hallmark (glycolysis, mTORC1 signaling). Gene set variation analysis (GSVA) was performed on each sample to obtain an enrichment score (ES) using the GSVA R package [42].

### 2.9. Statistical Analysis

Statistical analysis was performed using a 2-tailed Student t test (when comparing two groups) and One-Way ANOVA (three or more groups). Data are presented as means ± standard deviation and calculated using the software package GraphPad Prism version 8.0.1 for Windows, GraphPad Software, San Diego, California USA. RT-qRT-PCR expression data was graphed and analyzed directly in CFX Maestro 1.1 software, version 4.1.2433.1219 (Bio-Rad Laboratories). *p* values < 0.05 were considered as statistically significant. For all figures, *p* values were expressed according to GraphPad 8 NEJM *p*-value style.: *p* > 0.05 (ns); *p* < 0.05 (*); *p* < 0.01 (**); *p* < 0.001 (***).

## 3. Results

### 3.1. GBM can be Stratified into Glycolytic and Oxidative Phenotypes

Molecular heterogeneity is a key feature of GBM, with clinical and therapeutic repercussions. To better understand if the vast molecular landscape of GBM could be reduced into a manageable number of metabolic categories, we explored the TCGA expression databases using a GSVA approach. Filtering for canonical gene sets of glycolytic and oxidative pathways, Warburg-like phenotypes were enriched in the mesenchymal subgroup, whereas functional mitochondrial metabolism predominated in healthy tissues (Figure 1a). Between these two extremes, however, we still encountered ample metabolic heterogeneity. Clinically, 18F-FDG PET is valuable for staging and detection of recurrence, but not necessarily to guide treatments. Without further stratification, GBM can be identified as a malignancy with high glucose uptake or low glucose uptake (Figure 1b). Nevertheless, common standardized procedures such as 18F-FDG PET do not allow for differentiation between high glucose uptake due to increased aerobic glycolysis or OXPHOS, or low glucose uptake due to compensatory glutaminolysis, necrosis or quiescent metabolic phenotypes.

Subsequently, to examine the differences in bioenergetic metabolism in phenotypically and molecularly distinct gliomas in vitro, we determined OCR and ECAR of our set of GSCs and U87MG (Figure 1c). We can observe that, in basal conditions, GBM27 and U87MG are close to a 1:1 ratio of glycolytic/oxidative metabolism, whereas GBM18 and GBM38 have a strong preference towards a glycolytic phenotype. Furthermore, U87MG, a typically Warburg-like cell line [30], also exhibited a relative elevation in OCR-linked ATP production rate (up to 43.57% of total ATP). As shown in Figure 1d, GBM27 demonstrated high mitochondrial ATP production as well as lower glycolytic ATP production when compared to GSCs GBM18 and GBM38; as much as 50% of its bioenergetic needs were met by OCR-linked ATP production. GBM18 and, especially, GBM38, relied predominantly on glycolytic metabolism (Warburg effect). In GBM27, OCR and ECAR fluctuated slightly between sets of biological experiments, indicating a range of metabolic flexibility: further investigation into metabolite allocation for energy production would be necessary to fully characterize this adaptive capacity. Our data indicate that even under the same cell culture conditions, distinct molecular characteristics of GSCs can in fact produce unique metabolic phenotypes. Interestingly, the global metabolic activity of U87MG is actually lower than GSCs.

Taken together, our data suggest that a high degree of metabolic variability is present between our GSCs, and their ATP production rates are faster than those of U87MG. GSCs and U87MG maintain a stable, basal, metabolic profile, and seem to be able to shift, to some extent, between aerobic glycolysis and OXPHOS to meet their bioenergetic needs.

### 3.2. GSCs Display a Heterogeneous Pattern of Resistance to Metabolic Inhibitors

At the outset, in order to determine the optimal doses to be used in future experiments, we exposed our GSCs (GBM18, GBM27, GBM38) and the U87MG cell line to escalating concentrations of selected metabolic drugs. Inhibitory curves for all time points are presented in Appendix A.

After conducting these experiments, we ascertained maximum inhibitory effects and reliable trends in the viability data at 72 h; therefore, for every cell line, IC50 at 72 h was considered as the optimal inhibitory concentration.

As we can appreciate in Figure 2, GBM27 had the highest resistance to all metabolic treatments except for DON, where, in turn, GBM18 required the highest concentrations to achieve IC50. For MF, GBM18 and GBM38 were the most sensitive cell lines (10.66 ± 3.162 mM and 21.33 ± 7.08 mM, respectively) and GBM27 the most resistant (77.41 ± 34.02 mM). U87MG revealed an intermediate resistance (42.51 ± 2.742 mM). For DCA, GBM38 required the lowest concentrations (13.52 ± 5.235 mM) and GBM27 the highest (40.61 ± 7.400 mM). In this case, GBM18 (29.20 ± 5.627 mM) and U87MG (27.10 ± 0.955 mM) showed no statistically significant differences in IC50 concentrations. For SOD, all cell lines required relatively high in vitro concentrations to reach 50% growth inhibition; no statistical significance was reached between groups. Lastly, regarding glutaminolysis inhibition by DON, U87MG required the lowest IC50 DON dose (99.70 ± 14.82 μM), followed by GBM27 (198.4 ± 44.13 μM) and GBM38 (286.9 ± 103.2 μM), whereas GBM18 was the most resistant (1505 ± 625.4 μM). It should be noted, however, that a closer look at the growth inhibition curves for DON in GBM18 reveals a cytostatic “threshold” around the IC50 value regardless of the dose, suggesting a non-linear inhibitory slope (Appendix A). Therefore, the IC50 provided is a statistical approximation owed to the resistance against the drug, but we should not always assume a linear correlation between dose and effect; this will become especially relevant in subsequent combinatory studies.

In summary, GBM18 was the most sensitive to MF, GBM38 to DCA and U87MG to DON; on the other hand, GBM27 was the most resistant to MF and DCA, while GBM18 required the highest doses of DON. A very high resistance towards SOD, as well as low variability in responses, was observed across all cell lines, so this drug was discarded from further assays.

### 3.3. Differences of Target Enzymes across Cell Lines Predicts Responses to Metabolic Inhibitors

To further investigate the relative sensitivity/resistance profiles of each cell line to our selection of metabolic drugs, we aimed to evaluate their basal genetic expression profiles (Figure 3a).

MF acts through inhibition of the electron transport chain (ETC) complex I, increasing the ADP/ATP ratio, but its primary downstream target is the activation of AMPK (phosphorylation of Thr172 of AMPKα1), which ultimately leads to mammalian target of rapamycin (mTOR) inhibition. We therefore evaluated expression of PRKAA1 and PRKAA2 (together, catalytic subunits of AMPK) and the mTOR gene. Although PRKAA2 was not expressed in our samples, PRKAA1 was significantly upregulated in GBM27 (9-fold relative to control) and, at similar levels (approximately 3 to 4-fold), in GBM18, GBM38 and U87MG. We also found the highest relative expression levels of mTOR in GBM27 and GBM38, but differences did not reach statistical significance. Additionally, we analyzed the phosphorylation of AMPKα to investigate the biological effects of MF (Figure 3b). We observed strong phosphorylation of Thr172 AMPKα relative to control in GBM18 (60 min, 2 h, 6 h) and GBM38 (2 h). Cell lines GBM27 and U87MG did not phosphorylate AMPKα in the first 6 h, consistent with their need for higher concentrations of MF and slower responses against the drug.

Next, we studied DCA activity in our GBM cell lines. Altough we analyzed all PDK subunits, PDK2 and PDK4 were not expressed, with major differences detected primarily in PDK3 expression, exceptionally upregulated in GBM27, consistent with sensitivity profiles to DCA. At the protein level, the catalytic subunit PDH-E1α has three major phosphorylation sites, with site 1 (Ser-293) being the most frequent and efficient target, sufficient to fully inhibit PDH activity [59,60]. With inhibition of PDKs by DCA treatment, we detected rapid, visually discernable, de-phosphorylation of Ser-293 in all cell lines after 6 h of treatment with 72h-IC50 concentrations (Figure 3c and Appendix A). The expression of PDKs in our dataset could provide a predictive biomarker to explain differential responses to DCA.

When we evaluated GLS genes, our analysis revealed no detectable amplification of GLS2; therefore, we focused on GLS1 as a potential predictor for DON’s antiproliferative effects. Relativized to epilepsy control, GLS1 was significantly upregulated in GBM18, neutral in GBM27/U87MG and downregulated in GBM38. Higher GLS1 expression correlated with the relative resistance against DON in GBM18, but comparatively lower expression in GBM38 was not associated with lower doses.

### 3.4. Doses of Metabolic Inhibitors and Radiomimetic Bleomycin Corresponding to Warburg-Like Phenotypes Spare Viability of Non-Tumoral hMSCs

After completing this set of experiments, we questioned whether our cell lines would respond favorably to bleomycin, a radiomimetic/DNA-targeting drug.

The mechanism of action and IC50-72h concentrations for bleomycin are presented in Figure 4a. To substantiate our following combinatory studies, we first performed exploratory MTS assays to determine optimal concentrations of bleomycin for each cell line: we observed relative resistance in GBM18 and GBM27, whereas GBM38 and U87MG were equally sensitive to the drug (Appendix A).

As with any other form of treatment, the success of metabolic therapies could be limited by toxicity to healthy cells. Therefore, we investigated whether all IC50 concentrations determined thus far could be a realistic goal in vitro, exploring their effects on non-tumoral hMSCs (Figure 4b). MF/DCA affected more than 60% of cells when using GBM27 IC50s (oxidative-like metabolic phenotype), but all other doses were very well tolerated (toxicity less than 20%). DCA 72h-IC50 from GBM38 actually slightly increased “cell viability” relative to control (as per metabolic activity measured by MTS assay). DON inhibited cell growth up to 35% at the highest dose (1505 µM), suggesting a saturation point after which increased dosages do not linearly correlate with antiproliferative effects. Bleomycin, in contrast, only affects up to 30% of hMSCs at the highest IC50, correlating well with slower proliferation rates. In summary, cell lines with predominantly Warburg-like phenotypes could be targeted with metabolic inhibitors without affecting normal stem cells, but oxidative-like phenotypes, such as GBM27, will require individualized, unique approaches.

### 3.5. Synergy between Bleomycin and Metabolic Inhibitors Helps to Overcome Dose-Limitng Toxicity in Predominantly Oxidative Metabolic Phenotypes

One way to solve the challenge of non-specific damage to healthy cells is to exploit coexisting weaknesses of tumoral cells in combinatory strategies. Since GBM27 doses of MF/DCA were also affecting the viability of normal cells, we explored the possibility of dose-reduction attributable to synergistic effects with radiomimetic bleomycin. After individually confirming the validity of each calculated 72h-IC50, we performed combinatory studies to determine the existence of synergy, additive or antagonistic effects. Using the Chou-Talalay theorem, the Combination Index (CI) and the Dose Reduction Index (DRI) were calculated for each drug combination.

As shown in Table 2, drug mixtures with CI  <  1 and DRI  >  1 at a fraction of affected cells (Fa)  = 0.6 were considered as the optimal cutoff to identify promising therapeutic combinations. Nevertheless, in constant ratio combinatory experiments, close attention needs to be paid to the full range of Fa and CI/DRI to evaluate synergy for any given combination/antiproliferative effect.

All final reports with complete datasets, including Median-Effect Plot, CI Plot, Logarithmic CI Plot, DRI, Isobologram and Sequential Deletion Analysis (SDA; confidence intervals for CI values), are included in Appendix A.

Our results describe a wide variety of combinatory effects depending on the cell subtype and Fa level. Figure 5a describes the combined effects of MF and bleomycin. GBM18 exhibits mostly additive effects (no synergy). GBM27, on the other hand, is a prototypical example of synergistic effects when affecting most tumoral cells (at high Fa values): in the CI index, Fa ≥ 0.75 has a CI < 0.4, indicative of strong synergism. Consequently, DRI is > 1 for both MF and bleomycin, with significant dose reductions at Fa ≥ 0.6, potentially reducing the toxicity of both agents. GBM38 is synergistic at Fa ≈ 0.5 but has a tendency towards antagonism at Fa > 0.75. GBM38 appears to have a threshold for both MF and bleomycin, where even small doses produce significant anti-proliferative effects, but further increases provide no additional benefit. Finally, the CI in U87MG is close to synergistic/additive up to Fa = 0.75, then turning antagonistic.

Next, Figure 5b describes the combinatory effects of DCA and bleomycin. For GBM18, the CI is > 1 (antagonistic) at any given Fa; despite this, DRI > 1 for bleomycin at high Fa levels indicates the possibility of dose-reduction. In GBM27, therapeutic effects are determined by Fa cutoff: close to Fa ≈ 0.75, CI is <1 (synergistic), with dose reduction predicted at this value. GBM38, on the other hand, represents a clear example of strong antagonism (CI values > 1.5 at any Fa level); consequently, the combination of drugs does not surpass the effects of bleomycin by itself. Even still, DRI suggests that bleomycin concentrations could be decreased at the expense of DCA. Lastly, up to Fa = 0.5, U87MG displays synergistic/additive effects, with antagonism prevailing thereafter; this translates to unfavorable dosing at Fa > 0.8 (DRI < 1 for both drugs).

To conclude, Figure 5c details the interaction between DON and bleomycin. For GBM18, combining the compounds presents a strong synergistic relationship at Fa close to 0.5; however, at higher Fa, this synergy is lost. Ascending DRI for DON indicates the potential of important dose-reduction. GBM27 benefits from additive effects close to Fa ≈ 0.5, and, as we move closer to Fa = 1, the combination becomes strongly synergistic; this too would allow for dose reduction. In GBM38, the combined treatment has a similar threshold as in U87MG, as even one eighth of the concentrations significantly decreases proliferation: in these two cell lines, CI is <1 and DRI > 1 at Fa levels > 0.5, making DON and bleomycin a very promising therapeutic combination.

In summary, for GBM27, synergism with radiomimetic bleomycin was observed for all metabolic inhibitors at therapeutically relevant Fa = 0.9. Combining metabolic inhibitors and bleomycin could be leveraged to reduce dosing requirements of oxidative-like GSC subtypes.

### 3.6. Bioenergetic Profiling after Metabolic Treatment Reveals Opportunities for Metabolic Priming in Surviving Cell Populations

Using Seahorse XF technology, we determined total ATP production and ratios of mitoATP/glycoATP production under metabolically treated conditions.

As shown in Figure 6a,b, normalized total ATP production was decreased in all treated cells, with the exception of DCA-treated GBM27; in this case, rather than a significant drop in total ATP, production shifted from glycolytic to oxidative metabolism, with total ATP rates remaining unaltered. Furthermore, consistent with the proposed biological function for each drug, we observed a reduction in mitoATP production and a shift toward glycolysis using MF. Even though IC50 values should affect all cell lines proportionately, we noticed that the reduction in total ATP production was less pronounced with lower doses; e.g., in GBM18 (lowest MF IC50 of 10.66 mM), mitochondrial ATP was almost completely abolished, but total glycolytic ATP dropped only marginally, indicating a surviving population of almost exclusively glycolytic cells. Mitochondrial ATP production was increased after treatment with DCA, especially in GBM27, a GSC with a clear preference towards oxidative metabolism under both basal and treated conditions. The XF Rate Index for DCA can provide an idea of the oxidative potential of each cell line: highest in GBM27, followed by GBM18 and lowest in GBM38/U87MG. In the case of DON, we can appreciate a notable reduction of total ATP for each calculated IC50 value. As previously stated, DON is a glutamine analog predominantly targeting GLS (inhibition of TCA cycle intermediaries from glutamine would be expected to reduce mitoATP, unless glutamine derived α-ketoglutarate was diverted towards biosynthesis or mitochondrial substrate-level phosphorylation rather than oxidized). Examining the XF ATP Rate Index, GBM18 and especially GBM27 shifted towards oxidative metabolism, whereas GBM38 and U87MG remained unaltered. In conclusion, DON did not consistently change the metabolic phenotype of surviving cells; interestingly, however, in U87MG, a characteristically glutaminolytic cell line, even small concentrations of DON (99.7 μM) were enough to drastically reduce total ATP production.

Normalized values of OCR/Proton Efflux Rate (PER) in real-time after each drug injection of the Seahorse XF protocol are provided in Appendix A. These kinetic graphs allow us to examine how previous metabolic treatments changed the basal metabolic state and acute responses to mitochondrial inhibitors included in the assay: oligomycin (complex V inhibitor, i.e., mitochondrial ATP synthesis) and rotenone/antimycin A (total inhibition of mitochondrial respiration, complex I and complex III, respectively). Glycolytic upregulation in response to partial and complete mitochondrial inhibition (Warburg phenotype) and residual non-mitochondrial OCR should be highlighted in GBM18 and GBM38.

In conclusion, pre-treatment with metabolic drugs could become a valuable tool to characterize the bioenergetic phenotypes of surviving, resistant fraction of cells, priming them for further treatments. To our knowledge, this is the first time chronic metabolic treatments (>72 h) with DCA and DON have been characterized in GSCs. Despite methodological differences, our results also confirm the bioenergetic changes after low-dose MF in GSCs [61] and DCA in established glioma cell lines [62,63]. Finally, oxidative GSCs such as GBM27, an abnormality in the predominantly glycolytic phenotypes of solid tumors, need to be fully recognized in order to improve metabolic therapies. Based on our results, sequential drug strategies targeting previously weakened ATP-generating pathways warrant further exploration.

## 4. Discussion

Cancer metabolism has recently regained interest as a promising therapeutic target, coinciding with the development of standardized techniques for real-time assessment of bioenergetics. Given high intratumoral molecular and metabolic heterogeneity, it is necessary to target glycolytic, glutaminolytic and oxidative phenotypes; for this purpose, primary GSCs cultured from human surgical samples are an excellent model [64].

In this paper, we describe how metabolic modulators (MF, DCA, DON) could be leveraged to inhibit GSCs proliferation, demonstrating that distinct, stable, metabolic phenotypes contrast in their response to these treatments. A comparison between untreated and metabolically treated GSCs populations is presented here for the first time, leading us to hypothesize that inhibiting metabolic pathways might not only kill malignant cells, but also turn them vulnerable to successive targeted treatments (“metabolic priming”).

Consistent with our results, GSCs have been characterized by ample metabolic heterogeneity, exhibiting both glycolytic/oxidative characteristics [29,65,66]. Surprisingly, GBM27 exhibited a highly oxidative phenotype, at odds with Warburg’s central hypothesis of dysfunctional mitochondria; the presence of such metabolic subpopulations will need to be addressed to prevent tumor recurrence after antiglycolytic therapy. Similarly, U87MG had been described numerous times as highly glycolytic (high basal ECAR); however, we and others also described simultaneous elevations in OCR [30,67,68,69,70].

Regarding our selection of metabolic inhibitors, GBM27 was the most resistant to all drugs, with the exception of DON. One of the reasons for this might be owed to GBM27’s characteristically oxidative metabolism (up to 50% oxidative) and slower proliferation rates. Metabolic flexibility may grant a survival advantage even if mitochondrial energy production is targeted via MF or glycolysis via DCA. GBM18 and GBM38, on the other hand, were significantly more sensitive to MF; this would be consistent with their Warburg-like metabolic phenotypes. At the protein level, our results are consistent with previous research, where 10 mM MF did not significantly increase phospho-Thr172 at 6 h in U87MG [71]. Next, we hypothesized whether upregulated or downregulated PDK expression could be useful in predicting responses to DCA. While PDK1 was upregulated with respect to non-tumoral controls, expression was similar across all cell lines. PDK3 expression was relatively downregulated in GBM38 and exceptionally upregulated in GBM27. This would be consistent with previous reports where the PDK3 subunit was the most resistant to inhibition by DCA (higher expression of PDK3 would therefore translate into higher doses of DCA) [72,73]. In contrast, PDK2 subunits are the most sensitive to DCA treatment, while PDK1 and PDK4 display intermediate sensitivity (*Ki* PDK2 < PDK1 ≃ PDK4 < PDK3). As PDK1 was equally distributed, and PDK2/PDK4 were undetectable in our samples, variances in dose-response profiles were likely related to differences in PDK3. SOD treatment was equally inefficacious in all tested cell lines, but we suspect this was partly related to poor cell membrane permeability of this compound [74,75,76]. Despite issues with potency, our GSCs appear as highly resistant to SOD, as evidenced by much lower reported IC50s in nasopharyngeal carcinoma and gastric cell lines [77,78]. Concerning DON, glutamine addiction is thought to play an important role in supporting cancer proliferation [79,80]. In our case, GBM18 was the most resistant to DON, while other GSCs and U87MG turned more vulnerable to glutaminase inhibition. Our results confirm the inhibition profile for U87MG/DON at 72 h recounted by Ohba et al., with a similarly flattened inhibitory curve but higher “theoretical” IC50 [81]. With DON, rather than linear, dose-dependent cytotoxicity, we noted a more cytostatic-like effect in GBM18, GBM38 and, to a lesser extent, GBM27. Higher relative mRNA expression of GLS1 was evidenced in GBM18, accompanied with, indeed, higher doses of DON for the same inhibitory effect; correlations between doses and GLS1 expression was not as clear in other cell lines.

The heterogeneity of cancer could also be managed by combinatorial strategies, focusing on multiple molecular pathways to achieve compounded effects. Even though stand-alone metabolic modulation could become a possibility in the future, a more realistic and approachable goal is to design safe and effective combinations together with well-established chemotherapeutics and radiotherapy. Here, we show that non-tumoral hMSCs are not affected by low/intermediate doses of MF, DCA, DON or bleomycin. Concentrations of MF and DCA corresponding to GBM27, however, decreased viability by more than half in hMSCs, revealing that oxidative cancer cells would require a different management strategy: combinatory therapies could be such an option.

To the extent of our knowledge, ours is the first report of the combinatorial effects in GSCs of the radiomimetic drug bleomycin together with MF, DCA and DON. Bleomycin acts by inducing single and double strand DNA breaks, causing apoptosis and cell cycle arrest in early G2 phase [44]; it is FDA-approved as a clinical prescription against lymphomas, squamous-cell carcinomas and germ-cell tumours, as well as brain cancer [45,82]. IC50 profiles for bleomycin revealed significant disparities among cell lines, with GBM38 and U87MG being more sensitive than GBM18 and GBM27. Direct comparisons with radiotherapy are scarce, however, as we only encounter combined evaluation in traditional glioblastoma cell lines (not GSCs) for MF [61] and DCA [69]. Discussing our results, we want to bring attention to the fact that GBM27, identified as distinctly “oxidative”, was potentially the most benefited by pairing bleomycin and metabolic modulators. Synergistic opportunities in GBM27 would allow for significant dose reduction, as individual IC50 concentrations for MF and DCA are higher than in all other cell lines, thus reducing toxicity in normal cells. One of the strengths of our combinatory approach is the experimental design in a constant drug ratio: this allows for a data-driven, more accurate and comprehensive description of synergy/antagonism using the Chou-Talalay method [83], without relying solely on computer simulations.

As far as we know, sustained metabolic changes in GSCs after extended treatment with DCA and DON had not been properly illustrated until now. In terms of OCR/ECAR, all metabolic modulators exerted biologically coherent responses, especially well-defined for MF and DCA. In the case of MF, as would be expected, surviving cells shifted to a primarily glycolytic behavior, with a very low percentage of mitoATP generation; these results are consistent with Seahorse XF assays in established glioma cell lines and colon cancer CSCs [61]. The metabolic effects of MF were mainly determined by dose-response profiles. Similarly, DCA induced bioenergetic changes consistent with the forced activation of the PDH complex, in congruence with previous reports [69,84]. Interestingly, when comparing total ATP production rates, DCA only reached a maximal attenuation of approximately 50% relative to control. Indeed, DCA is believed to act more as a “metabolic modulator”, not simply reducing cell proliferation, but also “shifting” the glycolytic/OXPHOS ratio via increased mitochondrial uptake of pyruvate [85,86]. Lastly, evaluating the effects of DON, we detected consistent ATP inhibition but heterogeneous metabolic adaptations. In previous reports, no response was observed to low concentrations of DON in U87MG [87]. Preclinical data shows that relatively high in vitro doses of DON are required in some cell lines, which might explain the high variability in antiproliferative effects [88,89,90,91]. After chronic treatment with DON, we observed either an unchanged (GBM38, U87MG) or increased (GBM18, GBM27) ATP Rate Index. The third alternative, a decrease in ATP Rate Index (enhanced Warburg effect), was not detected; in contrast, this shift was noted with MF in all cell lines. Therefore, after GLS inhibition, we can hypothesize that increased mitoATP cannot be originated from glutamine oxidation: this was apparent in GBM27, where inhibition of glutaminolysis led to preferential oxidation of glucose (indicative of metabolic flexibility). GBM18 displayed a similar trend, albeit less pronounced, whereas GBM38 and U87MG saw their total ATP production diminished without compensatory shifts in mito/glycoATP.

From a translational medicine perspective, avoiding the risk of toxicity associated with high concentrations of MF might be feasible, as even very low doses of this drug revealed antiproliferative effects and radiotherapy enhancement [92,93]. Furthermore, combinatory strategies would allow for meaningful dose reductions, as shown with improved efficacy in radioresistant stem cells [94]. Moreover, DCA could be applied synergistically to prevent MF-induced lactic acidosis [95,96,97]. Our results agree with previously published data, showing that concentrations of DCA in the range of ≥ 10 mM are required for consistent anti-proliferative responses [98,99,100,101]. Finally, the prospective of DON for GBM therapy is hindered by a polar structure and reactive moiety that significantly reduces its ability to cross the blood-brain barrier (BBB) [102,103]. Nevertheless, most GBM patients present with a disrupted BBB [104], and, where this might not be applicable, prodrugs and novel glutaminase inhibitors are in active development [31,105].

We believe the origin of cancer is complex, with involvement of both oncogenic signaling and metabolic reprogramming [106,107]. Future directions in this field might entail a more precise molecular classification of tumor biopsies, expanding the comparison between gene expression subtypes and their metabolic milieu, and imaging of patients for improved, tailor-made therapeutics. Stratification and metabolic analysis will be crucial to discover malignancies that could benefit from adjuvant anti-glycolytic therapy [108,109], specific pathway inhibition (e.g., mitochondrial, glutamine and FAO inhibitors [110,111,112]) and tumors where, due to molecular rewiring, glycolysis inhibition with compensatory fuels might even be contraindicated [25,113,114].

## 5. Conclusions

In summary, we show that molecularly distinct GSCs show a high degree of heterogeneity, both in basal metabolic phenotypes and in response to metabolic inhibitors. Subpopulations of GSCs presented both highly glycolytic and highly oxidative characteristics. Stratification of patients according to coexisting metabolic phenotypes could be advanced into clinical tools for improved metabolic therapies. In highly oxidative GSCs (e.g., GBM27), there is potential for synergistic effects with radiomimetic bleomycin. Inhibition of glutaminolysis (via DON) is also an attractive therapeutic target, as synergy was described regardless of basal metabolism. Application of metabolic drugs produces stable changes in the bioenergetic states of GSCs, which could be leveraged as a form of metabolic priming.

## Figures and Tables

**Figure 1 cells-10-00202-f001:**
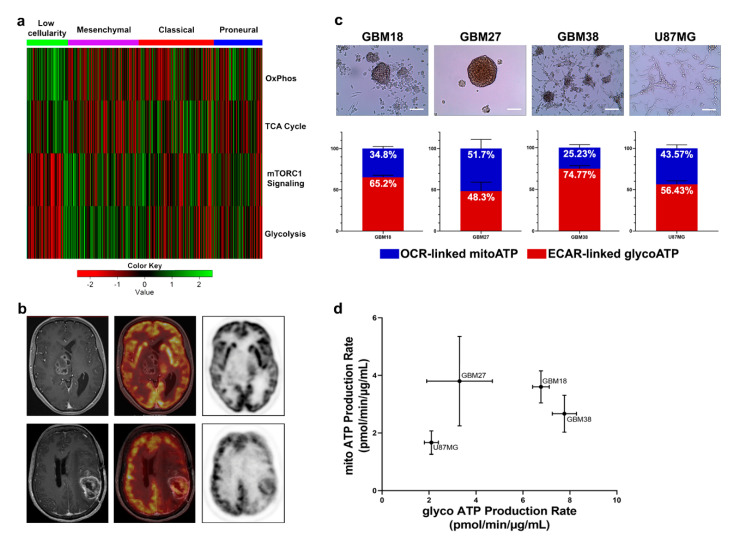
(**a**) Heatmap of the scaled enrichment score (ES) obtained by Gene Set Variation Analysis (GSVA) with the samples grouped by their gene expression subtype (proneural, classical or mesenchymal) including those with high content of non-tumor tissue (low cellularity). (**b**) Clinically, standard imaging techniques such as 18F-FDG PET-MRI can classify tumors into low glucose uptake (upper) and high glucose uptake (lower). Upper images: Right thalamic glioblastoma shows patchy contrast enhanced areas on 3DT1 (right side) and no uptake of 18F-FDG PET (medium and left side). Lower images: Parietal recurrent glioblastoma in the left hemisphere shows heterogeneous enhancement on axial three-dimensional T1-weighted imaging (3DT1) and extensive uptake of 18F-FDG PET (right and medium side), despite high uptake in surrounding normal brain tissue. (**c**) Representative optical microscopy images of cellular morphology. Scale bar = 100 µm. Under each cell line, average distribution of total ATP production from extracellular acidification rate (ECAR)-linked ATP production and oxygen consumption rate (OCR)-linked ATP production in basal (non-treated) conditions. (**d**) Seahorse XF Energetic Map. GBM18 and GBM38 clustered together as highly glycolytic-like cells. GBM27 displayed the highest variation in the metabolic profiles, with increased mitochondrial respiration, at a similar level to GBM18, but, in comparison, lower glycolysis. U87MG were not as metabolically active as glioma stem cells (GSCs). Data from three independent experiments, each with *n* = 3, normalized to total protein concentration (µg/mL).

**Figure 2 cells-10-00202-f002:**
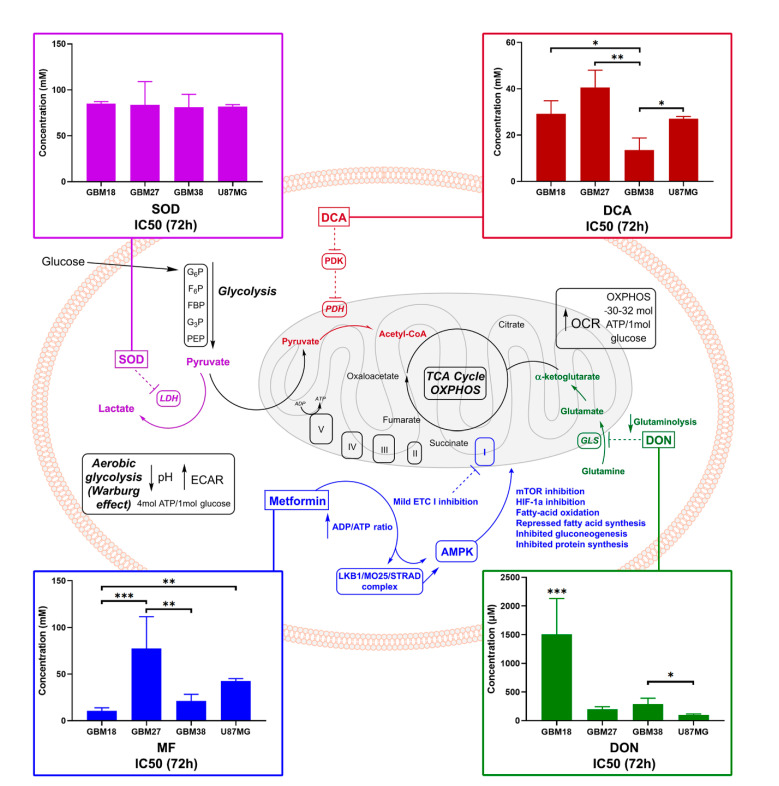
Cancer metabolism at a glance, with experimental in vitro IC50 values for selected metabolic inhibitors. Glucose enters cells and undergoes glycolysis, converted into pyruvate. Cancer cells can divert up to 85% of pyruvate to lactate, regardless of the presence of oxygen (Warburg effect, yielding two net ATP); an estimated 10% of pyruvate goes towards biosynthesis and 5% to OXPHOS [56]. In normal, non-tumoral cells, the majority of pyruvate undergoes OXPHOS (30–32 ATP molecules). To decrease the Warburg effect and facilitate oxidative metabolic reprogramming, PDKs can be inhibited by dichloroacetate (DCA), supporting the entry of pyruvate in the mitochondria, and LDH can be targeted via sodium oxamate (SOD). “Glutamine addiction” can be regulated by glutaminase inhibitors such as 6-diazo-5-oxo-L-norleucine (DON) [57]. Lastly, metformin hydrochloride (MF) has pleiotropic effects: ETC complex I inhibition leads to downstream signaling via AMPK and mTOR [58]. In color-matching boxes, we display concentrations required for 50% viability inhibition (IC50) after 72 h of treatment. One-way ANOVA statistical significance of three biological experiments was calculated with normalized raw fluorometric data; *p* < 0.05 *; *p* < 0.01 **; *p* < 0.001 ***. Abbreviations: ETC (electron transport chain), mTOR (mammalian target of rapamycin), PDK (pyruvate dehydrogenase kinase), PDH (pyruvate dehydrogenase), GLS (glutaminase), LDH (lactate dehydrogenase), GLS (glutaminase), ECAR (extracellular acidification rate), OCR (oxygen consumption rate).

**Figure 3 cells-10-00202-f003:**
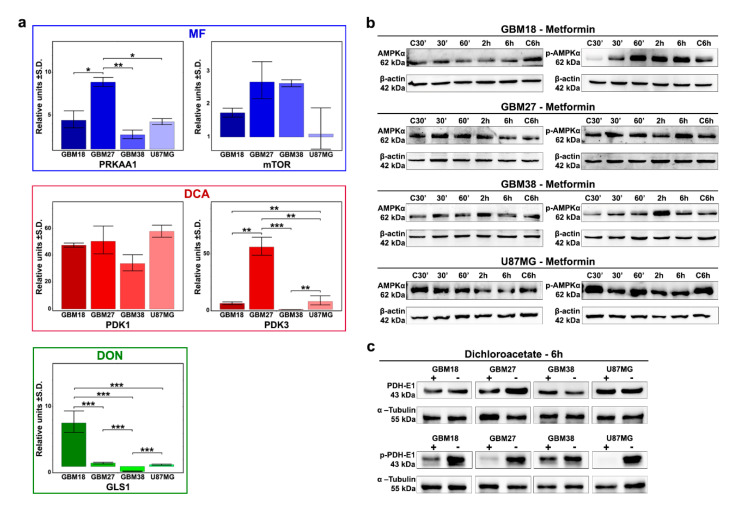
(**a**) Expression profiles of target enzymes for our selection of metabolic drugs under basal conditions determined by RT-qRT-PCR. Representative results from a minimum of two replicates (*n* = 2). One-way ANOVA with Tukey correction. *p* < 0.05 *; *p* < 0.01 **; *p* < 0.001 ***. (**b**) Western Blot analysis at 30 min, 60 min, 2 h, 6 h after MF 72h-IC50 treatment for AMPKα and phospho-Thr172 AMPKα. (**c**) Western Blot analysis after 6 h of treatment with respective DCA 72h-IC50 doses for phospho-Ser293 PDH-E1 and total PDH-E1.

**Figure 4 cells-10-00202-f004:**
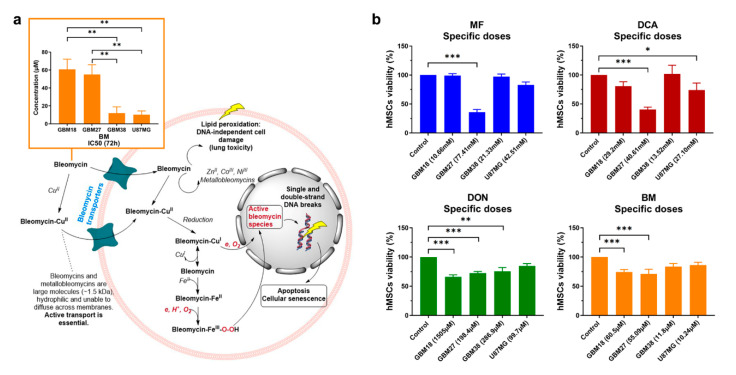
(**a**) IC50 values at 72 h and mechanism of action of radiomimetic drug bleomycin (BM). Bleomycin is a large molecule (~1.5 kDa) and cannot freely diffuse cell membranes; it is transported into cells either alone or as a bleomycin-Cu(II) complex, then reduced to bleomycin-Cu(I), which reacts with oxygen leading to DNA strand breaks. Successful chemotherapy with bleomycin is dependent on active transport; however, there is currently no consensus about the uptake mechanism or the transporters involved. Bleomycin-Cu(I) can also dissociate inside the cell to form bleomycin-Fe(II) complexes, transforming into «activated bleomycin species» resulting in DNA fragmentation and chromosomal aberrations. Complexes with zinc (II), iron (II) and cobalt (III) have also been characterized. Calculated IC50, as per inner salt (MTS) assay, with a minimum of two biological replicates. One-way ANOVA with Tukey correction, *p* < 0.05 *; *p* < 0.01 **; *p* < 0.001 ***. (**b**) Viability profiles relative to control mesenchymal stem cells (hMSCs) treated with all calculated 72h-IC50 doses (*n* = 2). One-way ANOVA with Dunn’s correction, *p* < 0.05 *; *p* < 0.01 **; *p* < 0.001 ***.

**Figure 5 cells-10-00202-f005:**
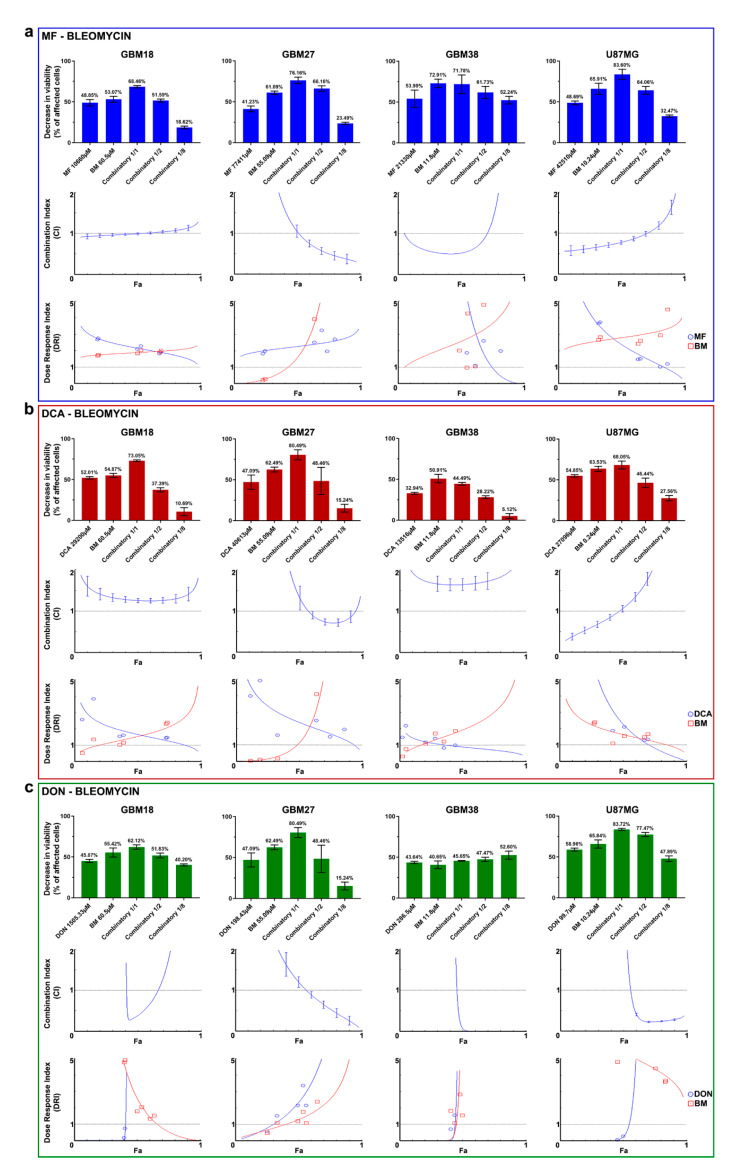
(**a**–**c**) Combinatory drug studies between metabolic drugs (MF, DCA, DON) and bleomycin (BM). From top to bottom, three graphs for each cell line comprehensively describe synergy/antagonism. First graph: Bar charts of decrease in viability (% of affected cells) relative to control. Second graph: Combination index (CI) is given as a function of the fraction of affected cells (Fa) by the drug combination with a continuous line. The central dashed line indicates a CI = 1. According to the Chou-Talalay’s Combination Index Theorem, CI = 0.9 to 1.1 indicates an additive effect. CI < 1 is indicative of synergism, whereas CI > 1 indicates antagonism. The vertical bars indicate 95% confidence intervals for CI values based on Sequential Deletion Analysis (SDA); in some cases, SDA values cannot be graphed in CompuSyn software, but they were always calculated and are available in Appendix A. Third graph: The Dose-Reduction Index (DRI) (also known as the Chou-Martin plot) signifies how many folds of dose-reduction for each drug at any given effect (Fa) are allowed in synergistic combination. In blue, metabolic drug DRI index; in red, bleomycin DRI index. DRI = 1 indicates no dose-reduction, DRI > 1 favorable dose-reduction and DRI < 1 no favorable dose-reduction. All experiments were performed in two biological replicates (*n* = 2).

**Figure 6 cells-10-00202-f006:**
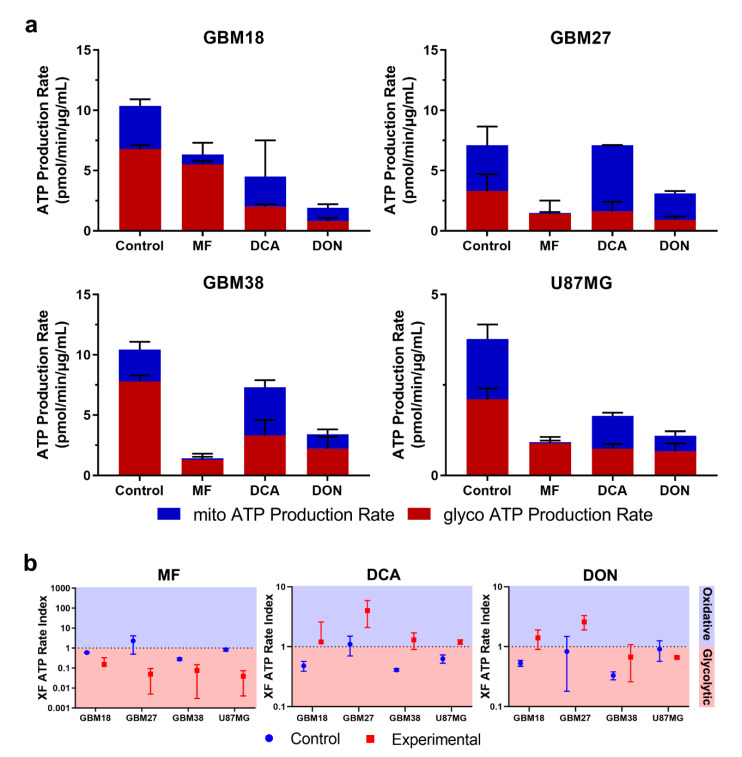
(**a**) Changes in metabolic phenotypes after IC50 treatment for 72 h with metformin (MF), dichloroacetate (DCA) and 6-Diazo-5-oxo-L-norleucine (DON). (**b**) XF ATP Rate Index for GSCs and U87MG. The ATP Rate Index is the ratio of the mitoATP Production Rate divided by glycoATP Production Rate, indicating higher oxidative or glycolytic bioenergetic profile.

**Table 1 cells-10-00202-t001:** Forward (FW) and reverse (RV) primers for real-time quantitative reverse transcription PCR (RT-qRT-PCR).

Name	5′-Sequence-3′
*β-actin* FW	TTCTACAATGAGCTGCGTGTG
*β-actin* RV	GGGGTGTTGAAGGTCTCAAA
*GAPDH* FW	TCCTCCACCTTTGACGCTG
*GAPDH* RV	ACCACCCTGTTGCTGTAGCC
*GLS1* FW	GCCCGCTTTGTGTGACTAAA
*GLS1* RV	CAGGGGTAAATAACGGCACA
*GLS2* FW	GCACTAAAGGCCACTGGAC
*GLS2* RV	CCAAGAGGCCACCACTACTG
*MTOR* FW	CTGACCGCTAGTAGGGAGGT
*MTOR* RV	AACATCCCAGAACCCTGCTG
*PDK1* FW	ATCCTCCTGCCTGAGTCTCT
*PDK1* RV	CAAATGCCAAGGACTGCTGT
*PDK2* FW	TGCCTACGACATGGCTAAGCTC
*PDK2* RV	GACGTAGACCATGTGAATCGGC
*PDK3* FW	TGGAAGGAGTGGGTACTGATGC
*PDK3* RV	GGATTGCTCCAATCATCGGCTTC
*PDK4* FW	AACTCGGGATGTTGGGGATT
*PDK4* RV	AGAGAAAAGCCCTTCCTACTGA
*PRKAA1* FW	GTCCAGGGCTTGTTCTATTCA
*PRKAA1* RV	ATGCTGCACTTAGAGACCCT
*PRKAA2* FW	TGGAACATTGTTACAGCAGGC
*PRKAA2* RV	AGCTCTTCTCCCGTGTCTTC

**Table 2 cells-10-00202-t002:** Summary of synergy/antagonism at an optimal Fa cutoff of = 0.6. All combinatory experiments were performed in two biological replicates (*n* = 2).

	MF + Bleomycin		DCA + Bleomycin		DON + Bleomycin	
Cell Line	Effect at Fa = 0.6	DRI at Fa = 0.6	Effect at Fa = 0.6	DRI at Fa = 0.6	Effect at Fa = 0.6	DRI at Fa = 0.6
GBM18	Additive	DRI > 1 for both	Antagonism	DRI > 1 for both	Synergism	DRI > 1 for both
GBM27	Synergism	DRI > 1 for both	Synergism	DRI > 1 for both	Synergism	DRI > 1 for both
GBM38	Synergism	DRI > 1 for both	Antagonism	DRI > 1 for bleomycin	Synergism	DRI > 1 for both
U87MG	Synergism	DRI > 1 for both	Antagonism	DRI > 1 for both	Synergism	DRI > 1 for both

## Data Availability

The data presented in this study is available in the Appendix A of this article. The results shown in the GSVA analysis are in part based upon data generated by the TCGA Research Network: https://www.cancer.gov/tcga.

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
