# Peer review of "Beyond the Warburg Effect: Oxidative and Glycolytic Phenotypes Coexist within the Metabolic Heterogeneity of Glioblastoma"

_cells, 2021, doi:10.3390/cells10020202_

Round 1

Reviewer 1 Report

The study is actual, accurately planned, performed, described and discussed. Thank you for providing better understanding about the role of the metabolic profile of glioblastoma in drug sensitivity. Well done.

Author Response

Dear Editor

Find included our reply to the Referees’ comments with respect to our original manuscript titled: “Beyond the Warburg effect: oxidative and glycolytic phenotypes coexist within the metabolic heterogeneity of glioblastoma” (cells-1079261). We are pleased the Referees find that “the manuscript is actual, accurately planned and well described” and we would like to thank them for their timely response and constructive suggestions. We believe that the addition of their comments has significantly improved the quality of the manuscript. In the following pages, we specify, point-by-point, the changes included in the new draft in line with yours and the Referees comments. We also indicate all the changes we have made in the manuscript with the track changes tool.

Reviewer #1

The study is actual, accurately planned, performed, described and discussed. Thank you for providing better understanding about the role of the metabolic profile of glioblastoma in drug sensitivity. Well done.

We would like to thank Reviewer #1 for his/her kind words and encouragement. Even though many battles lie ahead in the fight against brain cancer, we hope that a better understanding of the metabolic characteristics of this grave and mortal disease will open the door for improved therapies.

Reviewer 2 Report

I must say this paper is quite well done! There are some minor concerns I have, but those can be addressed quickly.

Concerns:

  1. The labelling of your figures, for example Figure 1D, is very small. The data would benefit from larger text. Figure 5 is exceedingly hard to understand, I think fewer acronyms would help here.
  2. Regard GBM38 sensitivity to DCA treatment, I'm struck by the fact that DCA treatment does not seem to dramatically inhibit PDH phosphorylation compared to other cell lines, yet is the most sensitive to DCA treatment. Why would GBM38, which doesn't express PDK3, and doesn't seem to inhibit PDH phosphorylation, be the most sensitive to DCA? It seems incredibly counter-intuitive. Can you provide a hypothesis/text to explain why this may be the case.
  3. Not really a concern, but the fact that the Warburg hypothesis doesn't hold is not surprising considering recent progress in the field (PMID: 32694689). I think this manuscript highlights this.
  4. It is a bit strange that the authors tested bleomycin in combination with metabolic inhibitors instead of testing standard of care therapies for the disease (temozolomide/radiation). What was the reason for this?

Author Response

Dear Editor

Find included our reply to the Referees’ comments with respect to our original manuscript titled: “Beyond the Warburg effect: oxidative and glycolytic phenotypes coexist within the metabolic heterogeneity of glioblastoma” (cells-1079261). We are pleased the Referees find that “the manuscript is actual, accurately planned and well described” and we would like to thank them for their timely response and constructive suggestions. We believe that the addition of their comments has significantly improved the quality of the manuscript. In the following pages, we specify, point-by-point, the changes included in the new draft in line with yours and the Referees comments. We also indicate all the changes we have made in the manuscript with the track changes tool.

Reviewer #2

1-The labelling of your figures, for example Figure 1D, is very small. The data would benefit from larger text. Figure 5 is exceedingly hard to understand, I think fewer acronyms would help here.

We agree with Reviewer #2 and take his/her input regarding text size of the Figures. Following his/her directions, we have modified Figure 1D to make the axis font bigger and clearer.

To further improve and enlarge the visual impact of the heatmap at the bottom of the figure we have also modified Figure 1A.

Similarly, we modified Figure 5 so that the graphs axis clearly states what are we referring to: “CI” = “Combinatory Index” and “DRI” = “Dose Response Index”, along with adding a “1” to where the discontinuous line in each CI graph is, to indicate the cut-off that describes synergy/additive/antagonism.

2-Regard GBM38 sensitivity to DCA treatment, I'm struck by the fact that DCA treatment does not seem to dramatically inhibit PDH phosphorylation compared to other cell lines, yet is the most sensitive to DCA treatment. Why would GBM38, which doesn't express PDK3, and doesn't seem to inhibit PDH phosphorylation, be the most sensitive to DCA? It seems incredibly counter-intuitive. Can you provide a hypothesis/text to explain why this may be the case.

We appreciate Reviewer #2 question regarding the potency of DCA as observed in the Western Blot. This fact was something we explored while performing our experiments but decided not to include in the main text as we thought it might distract from the clarity of Figure 3.

As correctly noted by Reviewer #2, Figure 3C displays total PDH-E1 and phospho-Ser293 PDH-E1, at 6h after the addition of treatment/control. The reason for this was to standardize our experiments and avoid confusion, because the dose of DCA determines the speed of its effects, and previously published studies using DCA situate cytotoxic effects and PDH phosphorylation depending on the dose, usually at 24h (see references below). When using IC50 concentrations, 6h was the optimal timepoint.

However, we agree that GBM38 deserves special attention and we also performed Western-Blots analysis with the following conditions:

  • Control cells collected after 30 minutes from adding fresh cell culture media (C30’)
  • DCA IC50 treated cells collected after 30 minutes, 60 minutes, 2h and 6h.

  • Control cells collected after 6h from fresh medium addition (C6h).

PDH regulation is a dynamic system that responds quickly to DCA treatment. In the case of GBM38, we observed stronger de-phosphorylation at 2h, even though at 6h it can still be visible. Therefore, what we see with GBM38 and DCA is that its effect is, in fact, faster than in other cell lines, with de-phosphorylation (activation) of PDH occurring at around 2h. At 6h (Figure 3C), dephosphorylation is visually discernable, but the magnitude is not as large as in other cell lines. We included these Western Blot images in Supplementary Material Figure S2 to help better understand the mechanism of action of DCA, thus modifying line 444 of the manuscript to indicate where they can be found (“Supplementary Figure S2”).

Similarly, regarding PDK subunit expression, we believe that a clearer explanation of the PDK sensitivity profiles to DCA might help to explain the unique profile of GBM38. In our discussion, we have indicated some information about it (lines 607-610), but, to improve our explanation, we have modified the text as follows: “Next, we hypothesized whether upregulated or downregulated PDK expression could be useful in predicting responses to DCA. While PDK1 was upregulated with respect to non-tumoral controls, expression was similar across all cell lines. PDK3 expression was relatively downregulated in GBM38 and exceptionally upregulated in GBM27. This would be consistent with previous reports where the PDK3 subunit was the most resistant to inhibition by DCA (higher expression of PDK3 would therefore translate into higher doses of DCA) [72,73]. In contrast, PDK2 subunits are the most sensitive to DCA treatment, while PDK1 and PDK4 display intermediate sensitivity (Ki PDK2 < PDK1≃ PDK4 < PDK3). As PDK1 was equally distributed, and PDK2/PDK4 were undetectable in our samples, variances in dose-response profiles were likely related to differences in PDK3.”

As indicated in the references, PDK3 is a subtype of PDK that requires higher amounts of DCA in order to be inhibited: if PDK3 expression is higher (as in GBM27), DCA concentrations will need to be increased for the same effects; and if it is relatively lower (as in GBM38), less DCA will be needed. We believe this might be the reason for the sensitivity profiles we detected in our case. We hope our explanation provided sufficient detail for a better understanding of our results.

Regarding PDH de-phosphorylation depending on the dose of DCA:

  • Ho, N., & Coomber, B. (2015). Pyruvate dehydrogenase kinase expression and metabolic changes following dichloroacetate exposure in anoxic human colorectal cancer cells. Experimental cell research, 331 1, 73-81.
  • Shen, H.; Decollogne, S.; Dilda, P.J.; Hau, E.; Chung, S.A.; Luk, P.P.; Hogg, P.J.; McDonald, K.L.J.J.o.E.; Research, C.C. Dual-targeting of aberrant glucose metabolism in glioblastoma. 2015, 34, 14.
  • Sun, R.C.; Fadia, M.; Dahlstrom, J.E.; Parish, C.R.; Board, P.G.; Blackburn, A.C.J.B.c.r.; treatment. Reversal of the glycolytic phenotype by dichloroacetate inhibits metastatic breast cancer cell growth in vitro and in vivo. 2010, 120, 253-260.
  • Heshe, D.; Hoogestraat, S.; Brauckmann, C.; Karst, U.; Boos, J.; Lanvers-Kaminsky, C.J.C.c.; pharmacology. Dichloroacetate metabolically targeted therapy defeats cytotoxicity of standard anticancer drugs. 2011, 67, 647-655.
  • Madhok, B.; Yeluri, S.; Perry, S.; Hughes, T.; Jayne, D.J.B.j.o.c. Dichloroacetate induces apoptosis and cell-cycle arrest in colorectal cancer cells. 2010, 102, 1746-1752.
  • Stockwin, L.H.; Yu, S.X.; Borgel, S.; Hancock, C.; Wolfe, T.L.; Phillips, L.R.; Hollingshead, M.G.; Newton, D.L.J.I.j.o.c. Sodium dichloroacetate selectively targets cells with defects in the mitochondrial ETC. 2010, 127, 2510-2519.

Regarding PDK subunits sensitivity profiles (as referenced in Discussion):

  • Baker, J.C.; Yan, X.; Peng, T.; Kasten, S.; Roche, T.E.J.J.o.B.C. Marked differences between two isoforms of human pyruvate dehydrogenase kinase. 2000, 275, 15773-15781.
  • Bowker-Kinley, M.M.; DAVIS, I.W.; Wu, P.; HARRIS, A.R.; POPOV, M.K.J.B.J. Evidence for existence of tissue-specific regulation of the mammalian pyruvate dehydrogenase complex. 1998, 329, 191-196.

3-Not really a concern, but the fact that the Warburg hypothesis doesn't hold is not surprising considering recent progress in the field (PMID: 32694689). I think this manuscript highlights this.

We thank Reviewer #2 for sharing this article with us and kind words regarding our contribution to the field. We think it’s important to acknowledge oxidative phenotypes to improve therapeutic outcomes, regardless of other roles mitochondrial dysfunction might play in the origin and progression of cancer. As elegantly articulated in the paper the reviewer links to: “impairments in oxidative metabolism may stimulate aerobic glycolysis in cancer, but, in general, aerobic glycolysis does not predict loss of oxidative metabolism”. We think that focusing on both issues, along with glutaminolytic tumours, intake of lactate (reverse Warburg effect), oxidation of fatty acids and even ketone bodies, might prove essential to design efficacious and highly personalized therapies for each patient. Standardization of imaging techniques and in vivo study of metabolism will need to “catch up”, as well as our understanding of coexisting metabolic phenotypes, to conquer these future therapeutic achievements.

4-It is a bit strange that the authors tested bleomycin in combination with metabolic inhibitors instead of testing standard of care therapies for the disease (temozolomide/radiation). What was the reason for this?

We think Reviewer #2 has a very good point, and we will try to explain why we focused on bleomycin as a radiomimetic drug, rather than temozolomide (TMZ) as standard of care.

First of all, the reasons we did not chose TMZ for combinatory studies were two-part: firstly, in previous work by our group and when performing exploratory TMZ viability assays, our GSCs cell lines displayed a high resistance towards TMZ (in the reference below or 46 in the manuscript, from our own team, we can see that 72h IC50 for TMZ in GBM18, GBM27 and GBM38 is approximately 1400-1700 μM). Similarly, our cell lines are extremely sensitive to DMSO and other organic solvents, making dilutions to such high concentrations of TMZ difficult. Secondly, apart from this technical reason, when exploring the literature, we observed that TMZ has already been evaluated in the full spectrum of clinical research: human trials, animal models and in vitro models of glioma, including GSCs, with moderate, dose-dependent effects (references below).

Secondly, another reason for choosing bleomycin instead of classic radiotherapy (irradiation of cell cultures) was simply technical, as we had limited access to irradiation equipment.  Therefore, because bleomycin works as a radiomimetic drug and is FDA approved for the treatments of many types of cancers (lymphomas, squamous-cell carcinomas and germ-cell tumors, as well as brain cancer, when standard of care fails), we thought it would be a good alternative. We believe that the results obtained by studying bleomycin can be taken either as a “proxy” of radiotherapy or they can stand on their own as a “chemotherapeutic agent”, as used in the clinic.

Evaluation of TMZ with MF and DCA:

  • Yu, Z.; Zhao, G.; Xie, G.; Zhao, L.; Chen, Y.; Yu, H.; Zhang, Z.; Li, C.; Li, Y.J.O. Metformin and temozolomide act synergistically to inhibit growth of glioma cells and glioma stem cells in vitro and in vivo. 2015, 6, 32930.
  • Dunbar, E.; Coats, B.; Shroads, A.; Langaee, T.; Lew, A.; Forder, J.; Shuster, J.; Wagner, D.; Stacpoole, P.J.I.n.d. Phase 1 trial of dichloroacetate (DCA) in adults with recurrent malignant brain tumors. 2014, 32, 452-464.
  • Yu, Z.; Zhao, G.; Li, P.; Li, Y.; Zhou, G.; Chen, Y.; Xie, G.J.O.l. Temozolomide in combination with metformin act synergistically to inhibit proliferation and expansion of glioma stem-like cells. 2016, 11, 2792-2800.
  • Gao-feng, X.; Mao-de, W.; Xiao-bin, B.; Wan-fu, X.; Rui-chun, L.; Chuan-kun, L.J.J.o.X.a.J.U. Combining dichloroacetate with temozolomide increases the antitumor efficacy of temozolomide by inhibiting HIF-1α and promoting p53 signaling pathway. 2014, 35.
  • Wicks, R.T.; Azadi, J.; Mangraviti, A.; Zhang, I.; Hwang, L.; Joshi, A.; Bow, H.; Hutt-Cabezas, M.; Martin, K.L.; Rudek, M.A.J.N.-o. Local delivery of cancer-cell glycolytic inhibitors in high-grade glioma. 2015, 17, 70-80.
  • Valtorta, S.; Dico, A.L.; Raccagni, I.; Gaglio, D.; Belloli, S.; Politi, L.S.; Martelli, C.; Diceglie, C.; Bonanomi, M.; Ercoli, G.J.O. Metformin and temozolomide, a synergic option to overcome resistance in glioblastoma multiforme models. 2017, 8, 113090.

Use of bleomycin in cancer:

  • Chen, J.; Stubbe, J.J.N.R.C. Bleomycins: towards better therapeutics. 2005, 5, 102-112.
  • Linnert, M.; Gehl, J.J.A.-c.d. Bleomycin treatment of brain tumors: an evaluation. 2009, 20, 157-164

Reviewer 3 Report

The authors characterized the basal bioenergetic metabolism and antiproliferative potential of metformin 22 (MF), dichloroacetate (DCA), sodium oxamate (SOD) and diazo-5-oxo-L-norleucine (DON) in 3 distinct glioma stem cells (GSCs) (GBM18, GBM27, GBM38), U87MG analyzing the Warburg effect.

The methodology used in this study is generally well described, and the results of the study encourage clinical trials to be undertaken using metabolism inhibitors.

  • A brief reference to the study referred to would be better for the immediate comprehensibility of the work (line 126);
  • It would be appropriate to reduce the introduction by better centering the point of the study
  • It would be appropriate to refer, possibly in a section of "further study", regarding the possibility of a comparative analysis between different GBM specimens with different characteristics (mesenchymal, proneural etc.)

Recommend minor revision

Author Response

Dear Editor

Find included our reply to the Referees’ comments with respect to our original manuscript titled: “Beyond the Warburg effect: oxidative and glycolytic phenotypes coexist within the metabolic heterogeneity of glioblastoma” (cells-1079261). We are pleased the Referees find that “the manuscript is actual, accurately planned and well described” and we would like to thank them for their timely response and constructive suggestions. We believe that the addition of their comments has significantly improved the quality of the manuscript. In the following pages, we specify, point-by-point, the changes included in the new draft in line with yours and the Referees comments. We also indicate all the changes we have made in the manuscript with the track changes tool.

Reviewer #3

The methodology used in this study is generally well described, and the results of the study encourage clinical trials to be undertaken using metabolism inhibitors.

1-A brief reference to the study referred to would be better for the immediate comprehensibility of the work (line 126)

We thank Reviewer #3 for his/her comment, and modified the line 126 accordingly, as follows: “GSCs were originally isolated from surgical human GBM specimens, as described by our group in [46]. The GSCs used in this study are characterized by distinct molecular and morphological features, differential drug sensitivity profiles and in vivo dissemination patterns that reflect the original tumors.”

We want to point to the fact that these cell lines have been extensively characterized by our team and employed in numerous publications since; detailed descriptions of neurosphere architecture, cell morphology, gene expression and stem cell markers, chromosomal imbalances and varied responses to drug treatments, as well as brain dissemination patterns in vivo, can be accessed in the reference provided. One of the reasons behind the current study was to complete characterization of these GSCs from a metabolic perspective. We hope this provides all necessary information and helps to clarify the origin of our cell lines.

GSCs characterization by our group:

  • García-Romero, N.; González-Tejedo, C.; Carrión-Navarro, J.; Esteban-Rubio, S.; Rackov, G.; Rodríguez-Fanjul, V.; Oliver-De La Cruz, J.; Prat-Acín, R.; Peris-Celda, M.; Blesa, D.J.O. Cancer stem cells from human glioblastoma resemble but do not mimic original tumors after in vitro passaging in serum-free media. 2016, 7, 65888.

Recent publications using our in vitro model of glioblastoma stem cells:

  • Rackov, G., Iegiani, G., Uribe, D., Quezada, C., Belda-Iniesta, C., Escobedo-Lucea, C., ... & Ayuso-Sacido, Á. (2020). Potential therapeutic effects of the neural stem cell-targeting antibody Nilo1 in patient-derived glioblastoma stem cells. Frontiers in oncology, 10, 1665.
  • García-Romero, N., Palacín-Aliana, I., Esteban-Rubio, S., Madurga, R., Rius-Rocabert, S., Carrión-Navarro, J., ... & Ayuso-Sacido, A. (2020). Newcastle Disease Virus (NDV) Oncolytic Activity in Human Glioma Tumors Is Dependent on CDKN2A-Type I IFN Gene Cluster Codeletion. Cells, 9(6), 1405.
  • García-Romero, N., Carrión-Navarro, J., Esteban-Rubio, S., Lázaro-Ibáñez, E., Peris-Celda, M., Alonso, M. M., ... & Ayuso-Sacido, A. (2017). DNA sequences within glioma-derived extracellular vesicles can cross the intact blood-brain barrier and be detected in peripheral blood of patients. Oncotarget, 8(1), 1416.
  • García-Romero, N., Palacín-Aliana, I., Madurga, R., Carrión-Navarro, J., Esteban-Rubio, S., Jiménez, B., ... & Ayuso-Sacido, A. (2020). Bevacizumab dose adjustment to improve clinical outcomes of glioblastoma. BMC medicine, 18(1), 1-16.

2-It would be appropriate to reduce the introduction by better centering the point of the study

As suggested by Reviewer #3, we made some text changes to the introduction to make it more focused. The list of changed lines can be consulted in the reviewed manuscript. We tried to summarize as much as possible and maintained only the information we feel relevant to help contextualize and understand the study.

3-It would be appropriate to refer, possibly in a section of "further study", regarding the possibility of a comparative analysis between different GBM specimens with different characteristics (mesenchymal, proneural etc.)

We included a brief section in Discussion (lines 711-714) referencing possible future directions, with the following paragraph:

“We believe the origin of cancer is complex, with involvement of both oncogenic signaling and metabolic reprogramming [106,107]. Future directions in this field might entail a more precise molecular classification of tumor biopsies, expanding the comparison between gene expression subtypes and their metabolic milieu, and imaging of patients for improved, tailor-made therapeutics.”

We thank all reviewers for their input, helpful suggestions and comments and hope our answers appropriately clarify our work.